# The Association between Prepartum Rumination Time, Activity and Dry Matter Intake and Subclinical Hypocalcemia and Hypomagnesemia in the First 3 Days Postpartum in Holstein Dairy Cows

**DOI:** 10.3390/ani13101621

**Published:** 2023-05-12

**Authors:** Mahmoud H. Emam, Elise Shepley, Mourad M. Mahmoud, Megan Ruch, Sobhy Elmaghawry, Wafaa Abdelrazik, Ahmed M. Abdelaal, Brian A. Crooker, Luciano S. Caixeta

**Affiliations:** 1Department of Veterinary Population Medicine, University of Minnesota, Saint Paul, MN 55108, USA; shepl044@umn.edu (E.S.); morad.mahmoud@vet.bsu.edu.eg (M.M.M.);; 2Department of Animal Medicine, Zagazig University, Zagazig 44511, Egypt; dr.sobhielmaghawry@gmail.com (S.E.); wmabdelrazek@vet.zu.edu.eg (W.A.); abdelaal79@yahoo.com (A.M.A.); 3Department of Animal Medicine, Beni-Suef University, Beni-Suef 62521, Egypt; 4Department of Animal Science, University of Minnesota, Saint Paul, MN 55108, USA; crook001@umn.edu

**Keywords:** activity behavior, postpartum disorders, precision technology, rumination behavior, transition cow management

## Abstract

**Simple Summary:**

There is increasing interest in the use of behavioral indicators to identify and even predict health disorders in dairy cattle, allowing for improved management and treatment strategies. The behavior changes such as the prepartum total daily rumination (TDR), total daily activity (TDA) and dry matter intake (DMI) have been suggested as predictors for subclinical hypocalcemia (SCH) and hypomagnesemia (HYM) postpartum. Our main objective was to evaluate the association between rate of change in rumination and activity behavior and DMI across the three days prior to calving with SCH and HYM status at D0 (day of calving) and D3 postpartum. Our results suggest that the rate of change in prepartum TDR, TDA and DMI is not an effective estimator of SCH and HYM status postpartum.

**Abstract:**

Changes in prepartum behaviors such as total daily rumination (TDR), total daily activity (TDA) and dry matter intake (DMI) have the potential to be used as early indicators for cows at risk for subclinical hypocalcemia (SCH) or hypomagnesemia (HYM) after calving. Our objective was to investigate associations between average daily rate of change in total daily rumination (ΔTDR), total daily activity (ΔTDA) and dry matter intake (ΔDMI) from −3 days prepartum to calving with SCH and HYM at D0 or D3 relative to calving. Prepartum TDR, TDA and DMI were measured in 64 Holstein dairy cows. Blood samples were taken at D0 and D3 post-calving for the measurement of total plasma Ca and Mg concentration. Linear regression models were used to analyze the association between ΔTDR, ΔTDA and ΔDMI and SCH and HYM at D0 and D3 relative to calving. Potential confounding variables were offered to the models and backwards selection was used to determine which covariates to retain. No significant differences in prepartum ΔTDR, ΔTDA or ΔDMI were found between cows with or without SCH and HYM at D0 and D3. Our results suggest that the change in TDR, TDA and DMI in the last 3 days prepartum are not effective predictors for cows that will have SCH or HYM in the first 3 days postpartum.

## 1. Introduction

The transition period from late gestation to early lactation is considered the most critical and challenging period in a dairy cow’s lactation cycle [1]. During this time, cows experience a significant increase in dietary requirements to support fetal growth and subsequent milk production [2,3], which leads to more stress during this vulnerable period [4]. Lactogenesis during early lactation rapidly increases mineral requirements, particularly for calcium which can increase by 65% [2]. Reduced concentrations of calcium (Ca), magnesium (Mg), potassium or phosphorus during the transition period leads to clinical or subclinical disorders which are defined as macro-mineral-related problems [5].

Hypocalcemia, particularly subclinical hypocalcemia (SCH), is a common metabolic disorder in dairy cows as most high-producing dairy cattle experience low blood Ca concentration during the periparturient period [6,7]. Subclinical hypocalcemia is defined by a blood Ca concentration below 8.8 mg/dL with an absence of additional clinical signs [8,9]. Despite the lack of clinical signs, SCH has been associated with decreased reproduction, production and health [6,10,11]. Recent research found that the risk of experiencing negative downstream effects of SCH is influenced by the time relative to calving that SCH occurs and the duration of time that Ca concentrations are low [9]. The same study determined that cows that suffer from delayed (i.e., SCH occurring after 1 DIM) or persistent (i.e., an extended period >4 DIM of low Ca) SCH had a greater risk of impaired health and performance than cows with transient SCH (i.e., SCH only in the first 24 h postpartum) or those that were normocalcemic [8,9]. Calcium and Mg metabolism are linked and any deficiency in one affects the other. Therefore, hypomagnesemia (HYM) represents a main risk factor for hypocalcemia [12,13]. A reduction of plasma Mg concentration affects the metabolism of Ca by decreasing the secretion of parathyroid hormone (PTH) and reducing tissue sensitivity to PTH [5,14]. Assessment of Mg status to avoid HYM during the transition phase is considered a monitoring tool for postpartum hypocalcemia [14].

Monitoring dairy cows during the transition period for early detection of subclinical diseases is important for the improvement of cow health and production [3,15,16]. Precision farming technology provides farmers with autonomous tools that can be used to monitor animal behavior and alert dairy staff to potential health issues [17] so they can improve management strategies [18,19]. These technologies may also have the potential to predict which cows are at risk for developing health disorders during the transition period. For instance, prepartum behaviors such as rumination time and total daily activity and feeding behavior have been associated with postpartum disorders [20,21], which demonstrates their potential as predictors of transition cow health and postpartum metabolic disorders [22,23,24]. Subclinical hypocalcemia reduces the strength and speed of smooth muscle contraction, and weakened rumen contractions adversely affects the rumination activity and rate of passage of digesta [25,26]. Additionally, SCH decreases dry matter intake (DMI) by dairy cows, and studies have identified a decline in the digestive function of cows suffering from SCH [9,25]. Feed intake provides the main source of Ca and Mg for cows in early lactation. Thus, reductions in DMI can negatively affect the blood concentration of these minerals [12,13].

There is limited information regarding the associations of prepartum behavioral patterns for rumination, activity and DMI and Ca and Mg status postpartum. Our primary objective was to evaluate associations between rate of change in rumination and activity behavior across the last 3 days prior to calving and SCH and HYM at D0 (day of calving) and D3. Our secondary objective was to investigate associations between change in DMI across the last 2 days prior to calving and SCH and HYM at D0 and D3. We hypothesized that cows with SCH or HYM on D0 and or D3 postpartum would have a smaller change in total daily rumination, total daily activity and DMI across the last three days prepartum than cows that did not experience SCH or HYM postpartum.

## 2. Materials and Methods

A case control study was conducted from November 2021 to July 2022. A total of 64 Holstein dairy cows were enrolled in this study (27 primiparous and 37 multiparous). Of the cows enrolled, 16 were unselected Holsteins (static low-merit Holstein; a University of Minnesota herd of genetically stable cows bred to maintain 1960s genetics; [27,28,29]) and 48 were contemporary Holsteins. Cows in this study were enrolled 3 weeks prior to their expected calving date to adapt to the prepartum diet and were followed until 6 DIM. Animals were housed in individual tie-stalls and were fed non-limiting amounts of prepartum or postpartum TMR (Table 1) according to NRC recommendations. Feed was delivered once daily, and cows were milked twice daily.

Blood samples were collected at D0 and D3 relative to calving from coccygeal vessels using a 20-gauge, 2.54 cm blood collection needle and vacuum tubes containing lithium-heparin (Vacutainer, Becton Dickinson). Within 1 h of collection, samples were centrifuged at 1500× *g* for 15 min at 4 °C for plasma separation and isolated plasma stored at −20 °C for later analysis. Total blood plasma Ca and Mg concentrations were measured in duplicate with a small-scale chemistry analyzer (CataChemWell-T, Catachem Inc., Oxford, CT, USA) using the Arsenazo III method for Ca with an enzymatic kit (C294-05, CataChem Inc., Oxford, CT, USA) (assay range: 0.36 to 17.2 mg/dL) and Xylidyl blue method for Mg with an enzymatic kit (C355-01, CataChem Inc., Oxford, CT, USA) (assay range: 0.85 to 8.5 mg/dL). Intra- and inter-assay coefficients of variation, respectively, were 2.7 and 3.6% for Ca and 3.4 and 5.3% for Mg.

Subclinical hypocalcemia was defined as plasma Ca ≤ 8.6 mg/dL and ≤7.8 mg/dL at D0 for primiparous and multiparous cows, respectively, and Ca ≤ 8.8 at D3 for both primiparous and multiparous cows [8,9]. Cows with a blood Ca at or below the SCH thresholds were categorized as SCH+ and those with blood Ca concentrations above these thresholds were considered normocalcemic and classified as SCH−. Hypomagnesemia was defined as a blood Mg < 1.8 mg/dL at D0 and at D3 for primiparous and multiparous cows [30]. Cows with blood Mg < 1.8 mg/dL were considered HYM+ and those with concentrations ≥ 1.8 mg/dL were considered to be in a state of normomagnesemia and classified as HYM−.

Total daily rumination (TDR) and total daily activity (TDA) were recorded continuously using an ear-tag accelerometer (CowManager, Agis Automatisering BV, Harmelen, The Netherlands). Data for TDR and TDA was output as total min/h spent ruminating or being active and reported as the total number of min/d, with rumination or activity presented as a 24 h interval from 00:00 to 23:59. Information regarding daily milk yield for the first 6 DIM was obtained from the farm software (DelPro™ Farm Manager 5.11, version 2022.02.04.006, DeLaval, Tumba, Sweden) and DMI from −21 to 14 DIM was obtained from the farm records (Feed Supervisor, Supervisor Systems, Dresser, WI, USA).

The predictors of interest for this study were changes in TDR (ΔTDR) and TDA (ΔTDA) from −3 days relative to calving to the day of calving (D0), inclusive, as D0 represent the nadir for TDR and TDA, and the change in DMI (ΔDMI) from −3 to −1 days relative to calving, as −1 day prepartum represents the nadir day for DMI. The values for these changes are reported as the slope generated from the linear regression of the daily measurements and day of measurement during the period of interest (−3 days relative to calving to D0 or −3 to −1 days relative to calving) for each cow. Associations between prepartum ΔTDR, ΔTDA and ΔDMI and SCH and HYM status at D0 and D3 relative to calving were evaluated using linear regression models. Separate models (*n* = 12) were created for each outcome (SCH or HYM status at D0 and D3) and predictor (ΔTDR, ΔTDA and ΔDMI) combination. A number of 11 cows had missing or incomplete rumination and activity data and were eliminated from their respective analyses. An additional 9 cows were missing D0 blood samples and were excluded from all D0 analyses. Two cows were missing DMI data from the 6 days prior to calving, 1 cow was missing baseline (−14 to −7 DIM) DMI data, and 1 cow was missing baseline TDA and TDR data—all of which were excluded from analyses offering these covariates.

Cow health status for SCH (SCH+ or SCH−) and HYM (HYM+ or HYM−) was forced into their respective models as a fixed effect. Covariates offered to the models with ΔTDR as the predictor were parity (1, 2 or 3+), genotype (1960s or current Holsteins genetics), baseline TDR (mean TDR from −14 to −7 days relative to calving), and DMI in the week prior to calving (mean DMI across days from −6 to −1 relative to calving) and postpartum DMI and milk yield (mean from 1 to 6 DIM for each). Covariates offered to the models with ΔTDA as the predictor were parity, genotype, baseline TDA (mean TDA from −14 to −7 days relative to calving), DMI in the week prior to calving and postpartum DMI and milk yield. Covariates offered to the models with ΔDMI as the predictor were parity, genotype, baseline DMI (mean DMI from −14 to −7 DIM), and postpartum DMI and milk yield. Backward selection was used to obtain all final models with possible covariates removed when the *p* value was >0.10. To ensure that variables in the model were not multicollinear in nature, collinearity diagnostics were run on all initial and final models using the olsrr package in R [31] with all variance inflation factors (VIFs) falling below 5.

Post hoc power calculations were conducted for each analysis. The calculated power is presented in Appendix A, with calculations based on group mean and SD, with group considered to be cows with a positive disorder status (i.e., having subclinical hypocalcemia or hypomagnesemia) or having a negative disorder status. Power for change in rumination (ΔTDR) and activity (ΔTDA) from −3 days prepartum to calving is based on an ability to detect a difference between groups of 20 min/d. Power calculations for the change in dry matter intake (ΔDMI) from −3 to −1 days prepartum is based on an ability to detect a difference of 1 kg/d. Power for analyses of predictor ΔTDR ranged from 19.3 to 25.5%, 39.4 to 54.1% power for ΔTDA, and 54.2 to 74.3% power for ΔDMI.

## 3. Results

### 3.1. Descriptive Results

Of the cows enrolled in this study, 27, 13 and 24 were parity 1, 2 or 3 or greater at enrollment, respectively. Average daily DMI from −14 to −7 days relative to calving was 9.7 ± 3.99 kg/d. Total daily rumination and TDA during the same baseline period were 458.9 ± 100.20 min/d and 343.8 ± 74.99 min/d, respectively. The average Ca concentrations of SCH− cows were 2.3 mg/dL and 2.6 mg/dL greater at D0 and D3, respectively, than those of SCH+, while average Mg concentrations in HYM− were 0.7 mg/dL and 0.8 mg/dL greater at D0 and D3, respectively, than HYM+ (Table 2). Appendix A presents the covariates offered to each model and those that were retained for each model used in the analyses as well as the total degrees of freedom for each analysis.

### 3.2. Calcium

The final model used to investigate the associations between prepartum ΔTDR and SCH status at D0 postpartum included the covariates baseline TDR and baseline DMI, while the model for ΔTDR and SCH status at D3 included DMI in the week prior to calving and postpartum milk yield as covariates. The final models used to investigate the association between prepartum ΔTDA and SCH status at both D0 and D3 postpartum included DMI in the week prior to calving as a covariate. The final models used to evaluate the association between prepartum ΔDMI and SCH status at D0 and at D3 included baseline prepartum DMI as a covariate.

No differences in prepartum ΔTDR were detected between SCH+ and SCH− cows at D0 (*p* = 0.80) or D3 (*p* = 0.60; Figure 1). The ΔTDR for SCH+ and SCH− cows had a least-square mean of −44.9 min/d (95% CI: −63.2, −26.0) and −49.8 min/d (95% CI: −86.2, −13.4) at D0, −47.4 min/d (95% CI: −65.1, −29.6) and −39.5 min/d (95% CI: −62.9, −16.1) at D3, respectively. Similarly, the ΔTDA did not differ between SCH+ cows (44.5 min/d; 95% CI: 30.7, 58.3 and 46.5 min/d; 95% CI: 33.2, 59.8), SCH− cows (41.2 min/d; 95% CI: 14.4, 68.1 and 44.3 min/d; 95% CI: 26.8, 61.8) at D0 and D3, respectively. These differences were not significant at D0 (*p* = 0.83) or at D3 (*p* = 0.84; Figure 2). The SCH status at D0 (*p* = 0.72) or at D3 (*p* = 0.57) had no association with the change in DMI prepartum (Figure 3), with DMI decreasing from −3 to −1 days relative to calving at a similar rate regardless of SCH status.

### 3.3. Magnesium

The final model investigating the association between ΔTDR and HYM status at D0 included baseline TDR, DMI in the week prior to calving and milk yield as covariates. Baseline TDR and baseline DMI were retained in the model for HYM status at D3. The final model used to determine differences in ΔTDA in HYM status at D0 included baseline DMI as a covariate while the model for D3 retained baseline TDA, DMI in the week prior to calving and genotype as covariates. The final model investigating the association between ΔDMI prepartum and HYM status at D0 retained no covariates and the D3 model retained baseline DMI and genotype.

No differences between HYM status and ΔTDR were detected at D0 (*p* = 0.37) or at D3 (*p* = 0.49; Figure 4). Interestingly, where the ΔTDR decrease was smaller in HYM+ cows (least-square mean of −39.4 min/d, 95% CI: −60.9, −17.9) compared to HYM− cows (−54.4 min/d, 95% CI: −79.6, −29.1), the opposite was true for HYM status at D3, where the TDR decrease prepartum was larger for HYM+ cows (−47.5 min/d, 95% CI: −63.9, −31.1) than for HYM− cows (−36.7 min/d, 95% CI: −63.0, −10.5). While no significant differences were observed between HYM+ and HYM− cows at D0 (*p* = 0.69) or at D3 (*p* = 0.87; Figure 5), the least-square means of ΔTDA of HYM+ cows in both models were (D0 = 45.9 min/d, 95% CI: 29.7, 62.0; D3 = 46.3 min/d, 95% CI: 33.9, 58.6) and HYM− cows (D0 = 41.0 min/d, 95% CI: 22.0, 60.0; D3 = 44.4 min/d, 95% CI: 24.7, 64.1). As with the results for SCH status, neither HYM status at D0 (*p* = 0.89) nor at D3 (*p* = 0.70) was associated with ΔDMI prepartum (Figure 6). In addition, the differences in the rate of decrease in DMI prior to calving in both HYM status groups were small (<0.7 kg/d).

## 4. Discussion

Our study was exploratory in nature and sought to investigate associations between rate of change of prepartum rumination and total daily activity behaviors from −3 days prepartum to day of calving, inclusive, and SCH and HYM status at the day of calving (D0) and D3 after calving. We also sought to investigate associations between rate of change in DMI from −3 to −1 days prepartum and SCH and HYM status at D0 and D3. We did not find any association between these behaviors prepartum and postpartum mineral status at D0 or at D3.

Investigation of the behavioral patterns of transition cows is critical for producers to improve treatment protocols, managemental strategies and overall production [32,33,34]. Rumination behavior has been recommended as a predictor for rumen activity and general health condition. It can especially serve as a good indicator for disease detection during the transition period [23,35,36]. Indeed, studies have reported that prepartum rumination activity is a strong predictor of hypocalcemia status [37,38]; however, these studies evaluated more cows (*n* = 210) and focused on detecting clinical hypocalcemia whereas we targeted subclinical cases. The lack of a difference in ΔTDR between SCH+ and SCH− cows at 0 and 3 DIM in our study suggests that SCH has less of an impact than clinical hypocalcemia on TDR. These results agree with those reported by Liboreiro et al. [21] where change in daily rumination 3 weeks prepartum did not differ between cows diagnosed as SCH or healthy at 3 days postpartum. Similarly, another study reported a weak correlation between rumination during the day prior to calving and plasma Ca concentration in the first 12 h after calving [26]. As with our study looking at more extended prepartum rumination patterns, others also concluded that prepartum rumination in the 24 h prepartum was not a strong predictor of postpartum hypocalcemia development [26].

Activity and, in particular, deviations in normal activity patterns may also be useful predictors of health disorders such as SCH in dairy cattle. Calcium concentrations play a vital role in the skeletal muscle strength and density and smooth muscle activity [39]. Furthermore, low blood Ca concentrations can impair the cow’s ability to stand [12,30]. The prepartum ΔTDA values for our SCH− and SCH+ cows did not differ, and our findings agree with previous reports that found no association between SCH and prepartum activity [20]. In contrast, SCH cows were reported to have stood 2.6 h/d more than normocalcemic cows during the 24 h before calving and 2.7 h less than normocalcemic cows during the 24 h after calving [32]. While they did not measure cow discomfort, the authors postulated that discomfort might contribute to increased standing by SCH cows before parturition [32]. Although Jawor et al. [32] did not detect differences in the number of standing bouts, fewer lying and standing bouts occurred during the two weeks prior to calving in multiparous cows that were normocalcemic than in those that developed subclinical or clinical hypocalcemia [40].

Voluntary DMI declines during the weeks prior to calving due to the stress and reduction in the rumen size [1,41]. Moreover, DMI steadily declines in the last 3 days prior to calving and reaches its nadir at calving [41]. We detected that DMI decreased from −3 to −1 days relative to calving at a similar rate in cows regardless of their postpartum SCH status at D0 or D3. Thus, the rate of decrease in prepartum DMI does not appear to influence postpartum SCH status. A similar lack of associations between prepartum DMI and SCH dynamics postpartum were found in a previous study, with the authors concluding that DMI may be a poor predictor of the Ca status postpartum [9]. Our findings were further confirmed by another study, which found similar patterns in prepartum DMI in cows that would remain normocalcemic and cows that developed SCH after calving [26]. In contrast, another report by Jawor et al. [32] reported greater DMI during the last 2 weeks prior to calving in cows subsequently diagnosed as SCH during the first 24 h postpartum than in normocalcemic cows. Additional investigations of the mechanisms responsible for inducing SCH are needed before reliable prepartum predictors can be identified.

Magnesium plays a vital role in the absorption of Ca, and HYM is a strong risk factor for SCH [30]. Thus, the similarity of our associations between prepartum behaviors and postpartum HYM and those for postpartum SCH are not surprising. We did not detect any significant differences in ΔTDR from −3 days prepartum to the day calving, inclusive, nor ΔDMI from −3 to −1 days prepartum, inclusive, between HYM− and HYM+ cows. Limited information is available regarding the relationship between HYM and rumination behaviors. Cows have a minimal ability to regulate blood magnesium, and, as such, adequate dietary Mg is required to maintain Mg concentrations levels in the blood [5,30]. Smaller changes in prepartum DMI and rumination behavior may indicate that dietary Mg intake was maintained at adequate levels leading up to calving, which could explain why neither ΔTDR nor ΔDMI prepartum were associated with HYM status postpartum in the current study. This contrasts with results of a previous study, in which a positive correlation between Mg concentration and rumination time was found, with authors suggesting that this correlation may be due to the strong relationship between rumination time and DMI [42]. However, that study was focused on mid-lactation as opposed to our study, which targeted the −3 days prior to calving.

To the authors’ knowledge, no previous research has investigated the association between the prepartum daily activity and HYM status postpartum. Our work indicates prepartum TDA does not influence postpartum HYM status. Although HYM+ cows had a numerically higher ΔTDA than HYM− cows, the lower differences observed when comparing both groups at the different time points postpartum was not biologically relevant. Regardless of the HYM status, the TDA of the animals increased when approaching calving as signals for stress and discomfort of the cows increased [32].

Interestingly, all eight of the TDR and TDA models included baseline DMI as a covariate and although the baseline DMI covariate for the DMI models covered a different interval, three of these four models included baseline DMI as a covariate. Prepartum DMI represents the main source for prepartum Ca and Mg, which plays a vital role in the level of rumination and activity [5,30]. These results suggest that these prepartum DMI values (−6 to −1 DIM in the TDR and TDA analyses and −14 to −7 DIM for DMI analyses) should be considered when designing future studies investigating the associations between prepartum rumination, activity and DMI behaviors and SCH and HYM status.

Our study has some limitations, including a relatively small sample size. In addition, it is not surprising that the number of SCH+ and SCH− cows differed, as herd management goals include a focus on minimizing clinical and SCH. However, we identified more cows as SCH+ than as SCH− and these proportions are consistent with the high prevalence of SCH in the dairy industry. Furthermore, this higher number of SCH+ cows in our work may have resulted from the use of higher cut-off points for SCH taken from more recent research, in comparison to lower cut-off points used in older studies. The current study was underpowered and, as such, future research using a larger sample size may be warranted to determine if similar results are found with a sufficiently powered study. As such, our findings should be treated with caution, and future work should look to investigate the associations between prepartum behaviors and timing and duration of SCH occurrence (i.e., SCH dynamics [8,9,11]). These SCH dynamics may be more influential in the management of transition cow health and prevention of downstream outcomes than just considering SCH status, but we did not have enough cows to explore this in our current study. Investigating these differences across a larger sample size or considering other pre-calving metrics as predictors for postpartum SCH and HYM risk should be considered in future studies.

## 5. Conclusions

The change in TDR and TDA from −3 to 0 days relative to calving, inclusive, and change in DMI from −3 to −1 days relative to calving, inclusive, were not different for cows with different SCH and HYM statuses at D0 or at D3 postpartum. Our results indicate that the change of TDR, TDA and DMI immediately before calving might not be an effective estimator of SCH and HYM status postpartum. Improving our ability to detect and predict health disorders, such as SCH and HYM, in dairy cattle can allow for improved management and treatment strategies as well as a better understanding of the potential causal effects of transition cow disorders, warranting further investigation. Consideration in future studies for timing and duration of SCH and HYM and use of larger sample sizes may improve the inferences we can make regarding the predictive abilities of prepartum behaviors.

## Figures and Tables

**Figure 1 animals-13-01621-f001:**
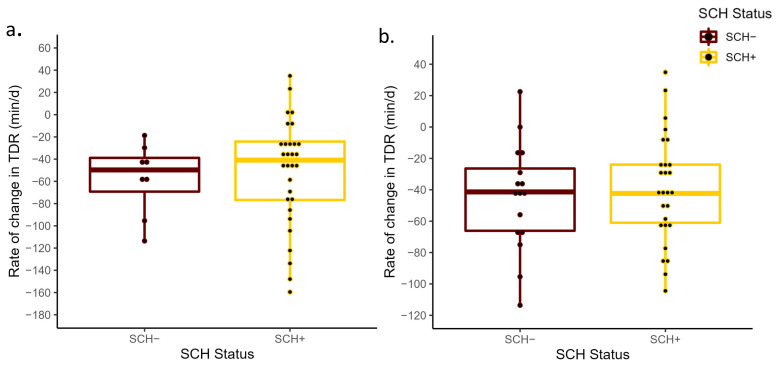
Mean rate of change in total daily rumination time from −3 days prepartum to calving (ΔTDR) for cows with subclinical hypocalcemia (SCH+; Ca concentrations ≤ 8.6 mg/dL (primiparous) and ≤7.8 mg/dL (multiparous) on D0 or Ca ≤ 8.8 mg/dL on D3) and those that were normocalcemic (SCH−) at D0 (*p* = 0.80; (**a**)) and D3 (*p* = 0.60; (**b**)) relative to calving. The ΔTDR was calculated based on the slope generated from the linear regression of the total daily rumination measurements and day of measurement from −3 days prepartum to calving.

**Figure 2 animals-13-01621-f002:**
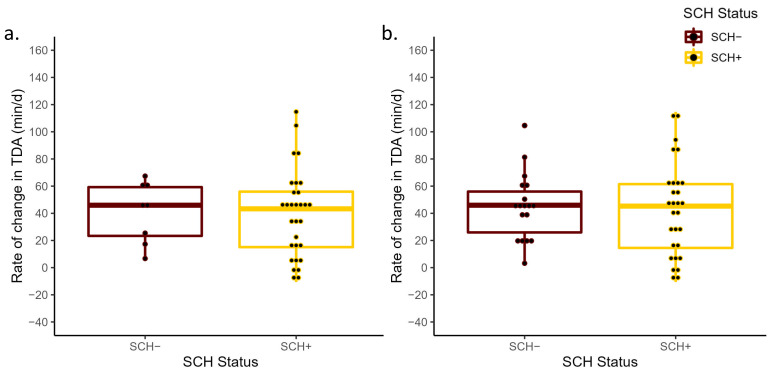
Mean rate of change in total daily activity time from −3 days prepartum to calving (ΔTDA) for cows with subclinical hypocalcemia (SCH+; Ca concentrations ≤ 8.6 mg/dL (primiparous) and ≤7.8 mg/dL (multiparous) on D0 or Ca ≤ 8.8 mg/dL on D3) and those that were normocalcemic (SCH−) at D0 (*p* = 0.83; (**a**)) and D3 (*p* = 0.84; (**b**)) relative to calving. The ΔTDA was calculated based on the slope generated from the linear regression of the total daily activity measurements and day of measurement from −3 days prepartum to calving.

**Figure 3 animals-13-01621-f003:**
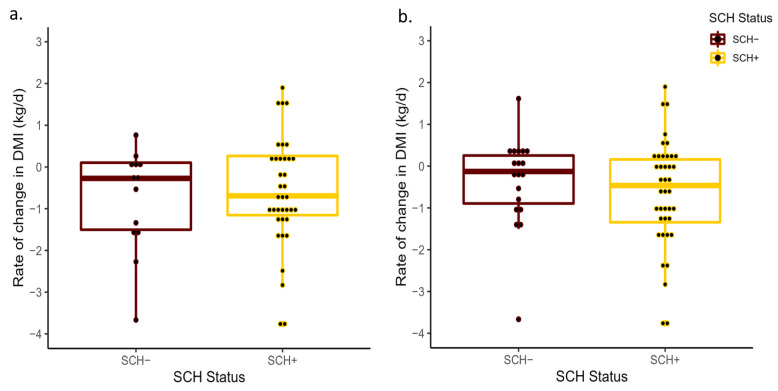
Mean rate of change in total daily dry matter intake from −3 to −1 days prepartum (ΔDMI) for cows with subclinical hypocalcemia (SCH+; Ca concentrations 8.6 mg/dL (primiparous) and ≤7.8 mg/dL (multiparous) on D0 or Ca 8.8 mg/dL on D3) and those that were normocalcemic (SCH−) at D0 (*p* = 0.72; (**a**)) and D3 (*p* = 0.57; (**b**)) relative to calving. The ΔDMI was calculated based on the slope generated from the linear regression of the total daily dry matter intake measurements and day of measurement from −3 to −1 days relative to calving.

**Figure 4 animals-13-01621-f004:**
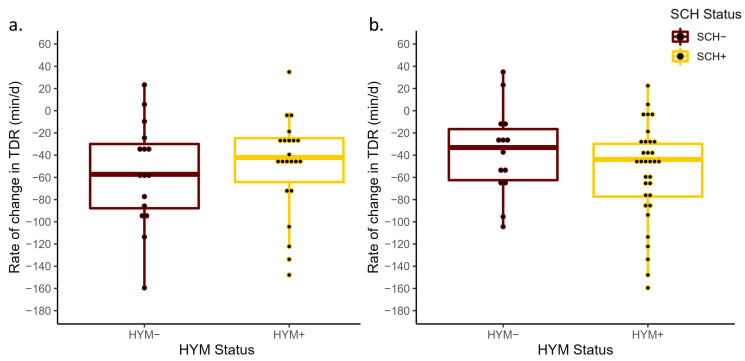
Mean rate of change in total daily rumination time from −3 days prepartum to calving (ΔTDR) for cows with hypomagnesemia (Mg concentrations < 1.8 mg/dL were classified as HYM+) and those with normal magnesium levels (HYM−) at D0 (*p* = 0.37; (**a**)) and D3 (*p* = 0.49; (**b**)) relative to calving. The ΔTDR was calculated based on the slope generated from the linear regression of the total daily rumination measurements and day of measurement from −3 days prepartum to calving.

**Figure 5 animals-13-01621-f005:**
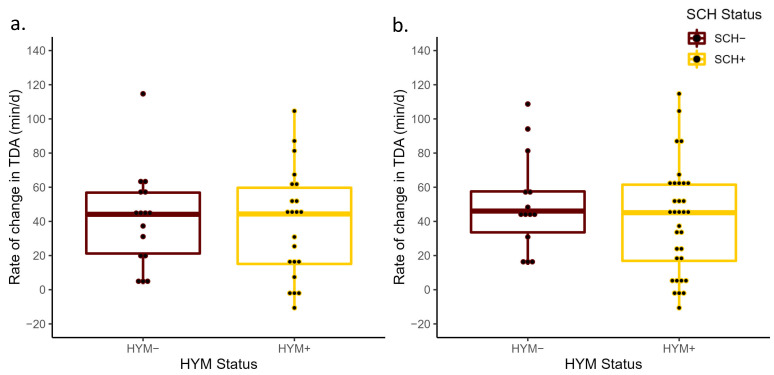
Mean rate of change in total daily activity time from −3 days prepartum to calving (ΔTDA) for cows with hypomagnesemia (Mg concentrations < 1.8 mg/dL were classified as HYM+) and those with normal magnesium levels (HYM−) at D0 (*p* = 0.70; (**a**)) and D3 (*p* = 0.87; (**b**)) relative to calving. The ΔTDA was calculated based on the slope generated from the linear regression of the total daily activity measurements and day of measurement from −3 days prepartum to calving.

**Figure 6 animals-13-01621-f006:**
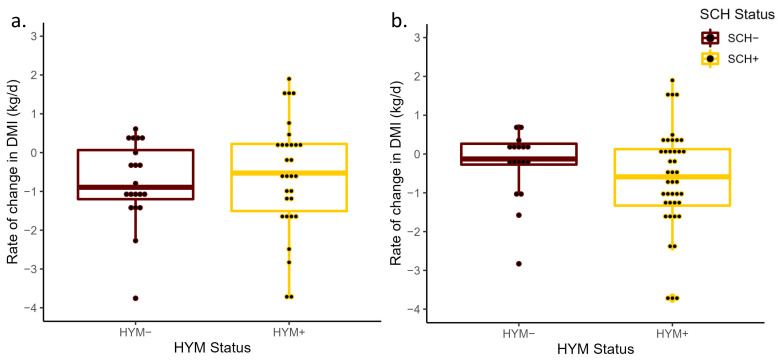
Mean rate of change in dry matter intake from −3 to −1 days prepartum (ΔDMI) for cows with hypomagnesemia (Mg concentrations < 1.8 mg/dL were classified as HYM+) and those with normal magnesium levels (HYM−) at D0 (*p* = 0.89; (**a**)) and D3 (*p* = 0.70; (**b**)) relative to calving. The ΔDMI was calculated based on the slope generated from the linear regression of the total daily dry matter intake measurements and day of measurement from −3 to −1 days relative to calving.

**Table 1 animals-13-01621-t001:** Ingredient and nutrient content of diets fed pre- and postpartum.

Components	Prepartum Diet	Postpartum Diet
Ingredient, % of DM		
Corn gluten	2.7	4.8
Corn silage	46.6	37.2
Grass hay	29.2	--
MegAnion premix ^1^	21.5	--
Alfalfa hay	--	10.8
Milk cow protein mix ^2^	--	14.5
QLF commercial dairy mix ^3^	--	5.0
Cottonseed, fuzzy	--	5.3
Corn, extra fine rolled	--	21.8
Energy Booster 100 ^4^	--	0.6
Nutrient content ^5^, DM basis (±SD)		
DM, %	51.1 ± 0.5	55.3 ± 0.9
CP, %	15.8 ± 0.2	16.3 ± 0.1
ADF, %	26.4 ± 0.5	16.6 ± 0.4
NDF, %	40.4 ± 0.9	25.9 ± 0.7
TDN, %	70.4 ± 0.2	77.2 ± 0.3
NE_L_, Mcal/kg	1.55 ± 0.01	1.72 ± 0.03
Ca, %	0.71 ± 0.01	0.88 ± 0.01
P, %	0.39 ± 0.01	0.38 ± 0.01
Mg, %	0.51 ± 0.00	0.36 ± 0.01
K,%	1.34 ± 0.01	1.43 ± 0.01
Na, %	0.07 ± 0.01	0.50 ± 0.00
Zn, ppm	75.9 ± 0.3	85.0 ± 0.2
Mn, ppm	69.3 ± 0.1	66.9 ± 0.07
S, %	0.28 ± 0.00	0.28 ± 0.01
Cl ion, %	1.35 ± 0.02	0.59 ± 0.01
Monensin, g/ton	20.2 ± 0.03	14.4 ± 0.00
DCAD ^6^, mEq/kg	−179.3 ± 1.3	242.9 ± 3.76

^1^ Bio-Sel Dry Cow 1000-Rum (Vita Plus Corporation, Madison, WI, USA), 2.46%; vitamin E-20,000, 0.05%; monocalcium phosphate, 21%; phosphorus, 0.05%; blood meal, 2.46%; calcium carbonate, 3.14%; canola meal, 21.61%; ReaShure Choline (Balchem Corporation, Montvale, NJ, USA), 1.91%; fat, 0.08%; magnesium oxide, 54%, 1.09%; soy hulls, 28.59%; soybean meal, 47%; protein, 21.61%; Yeast Original XPC (Diamond V, Cedar Rapids, IA, USA), 0.04%; MegAnion (Origination O2D, Maplewood, MN, USA), 14.63%. ^2^ Extra fine rolled corn, 30.67%; soybean meal 47%; protein, 17.00%; canola meal, 16.25%; Amino Plus (AG Processing, Inc., Omaha, NB, USA), 9.00%; blood meal, 6.00%; calcium carbonate, 5.25%; sodium bicarbonate, 5.00%; WR Elite Dairy Micro (Vita Plus Corporation, Madison, WI, USA), 2.60%; potassium carbonate, 2.00%; UltraMet (Vita Plus Corporation, Madison, WI, USA), 2.10%; sodium chloride, 2.05%; urea, 46%; N, 1.25%; Rumensin 90 (Elanco Animal Health, Greenfield, IN, USA), 0.03%. ^3^ Molasses-based liquid supplement of soluble sugars and protein developed from cane molasses, condensed whey, urea and sulfuric acid. The liquid supplement DM contained total sugars as invert 58.6%; CP, 11.8% with 38% NPN; fat, 0.5%; Ca, 1.16%; P, 0.42%; K, 4.82%; Mg, 0.47%; S, 0.95%; Na, 0.47%; Cl, 2.86%; Quality Liquid Feeds, Dodgeville, WI. ^4^ Hydrolyzed animal and vegetable fat. Milk Specialties Global, Eden Prairie, MN. ^5^ Samples collected weekly and composited bi-monthly for chemical analyses. ^6^ Calculated using the equation [(mEq of Na + mEq of K) − (mEq of Cl + mEq of S)].

**Table 2 animals-13-01621-t002:** Concentrations of blood plasma calcium (Ca) and magnesium (Mg) at the day of calving (D0) and 3 days postpartum (D3), presented overall and by health status. Cows with Ca concentrations ≤8.6 mg/dL (primiparous) and ≤7.8 mg/dL (multiparous) on D0 or Ca ≤8.8 mg/dL on D3 were classified as SCH+, otherwise SCH−. Cows with Mg concentrations <1.8 mg/dL were classified as HYM+, otherwise HYM−. Values are presented as the mean (95% confidence interval).

Variable	N	D0	N	D3
Ca concentrations, mg/dL				
Overall		6.6 (6.2, 7.0)		7.9 (7.5, 8.4)
SCH+	38	6.1 (5.7, 6.4)	41	7.0 (6.7, 7.3)
SCH−	13	8.4 (8.1, 8.8)	19	9.6 (9.2, 10.0)
Mg concentrations, mg/dL				
Overall		1.7 (1.5, 1.8)		1.5 (1.4, 1.6)
HYM+	31	1.4 (1.2, 1.5)	43	1.3 (1.2, 1.4)
HYM−	20	2.1 (2.0, 2.3)	17	2.1 (2.0, 2.2)

## Data Availability

The data presented in this study are available on request from the corresponding author.

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
