# Peer review of "The Association between Prepartum Rumination Time, Activity and Dry Matter Intake and Subclinical Hypocalcemia and Hypomagnesemia in the First 3 Days Postpartum in Holstein Dairy Cows"

_animals, 2023, doi:10.3390/ani13101621_

Round 1
Reviewer 1 Report
Very interesting research and research needed by cattle breeders.
The main question addressed by the research is can subclinical hypocalcemia (SCH) or hypomagnesemia (HYMN) be predicted based on rumination intensity and pre-calving feed intake.
This is an original and important topic for practice.
It brings something new. You can see the difference, is not supported statistically due to the small number. They should increase the number of cows and include a minimum of 7 days before calving.
The conclusions are consistent with the evidence and arguments presented and they address the main question posed.
The references are appropriate.
Line 26: TDR, TDA, and DMI were measured … I don't know what the abbreviations mean; Table 1 MegAnion premix1 - 21.5% of dry matter ! is this not a mistake? Table 2 specify how many cows are involved. There are differences! No significant differences were obtained because too few cows were included.
Author Response
Dear reviewer
We would like to thank you for the time and effort you put into revising our manuscript. Please see the attachment file for our response.

Reviewer 2 Report
Thank you for submitting your paper on using prepartum behavior and intake to predict mineral status of dairy cattle postpartum. This topic is of growing interest and preliminary data will help to determine which variables to focus on when trying to understand disease or disorder detection.
Line 18: It is not clear if the rate of change is daily rate of change or across 3 days.
Abstract: indicate the sample size in the abstract that had hypocalcemia and hypomagnesemia, with only 51 cows, it seems unlikely that you have sufficient power
Line 26: it is unclear why using just D0 and D3 for this analysis if you are interested in mineral status as you state later in the introduction about the transient nature of mineral status in early lactation.
Line 29-30: it is unclear what you mean by models were adjusted for and the timeline that these, here you indicate up to 6 DIM but other places it is 3 DIM.
Line 30: unclear what is meant by genotype here
Line 39: I believe you are really focusing on the days immediately after calving for this work and therefore this first sentence could be more focused on that time frame rather than the three weeks postpartum.
Line 43-45: you do not discuss lipid mobilization or oxidative status anywhere else, this is not needed, just talk about mineral status here.
Line 54-59: As you point out hypocalcemia subgroups exist so it is unclear why you took blood only at D0 and D3 for this analysis.
Line 64-65: Assessment itself is not prevention.
Line 72-76: These two sentences are very similar and could be combined to avoid repetition.
Line 84: rate of change is unclear if you are saying daily rate of change or rate of change over several days. Also, I need justification for D3 sampling.
Line 93: how was 51 cows determined to be a sufficient sample size for this analysis?
Line 95: unclear what you mean by “unselected” Holsteins
Line 119: when were blood samples collected on D0 relative to calving or the time of day relative to feeding? If there is variation in time relative to calving this may really impact your D0 results. For D3 results, when relative to feeding?
Line 147: were other dates initially analyzed and -3 days relative to calving was used for a specific reason? More information on how this was decided would be helpful.
Line 159: Indicating the total number of animals that were used would be helpful for the reader here.
Line 160: how many were SCH+ and SCH-? Same for HYM, I am not seeing that explicitly stated anywhere.
Line 162: please define genotype as (1960 or current genetics) or similar so it is clear what this means to you.
Table 2: please indicate how many cows fall under each category in this table, it is also not clear if those that were below the thresholds on just one day if they are considered – for both days or just the day they were below the threshold. That may seem obvious but I started to question that as I was reading your paper.
Figure 1 - 6: Are these adjusted by anything (parity, genotype)? Please indicate how many are in each population in the figure description.
Line 272: I think you are very underpowered for this type of analysis; I may be wrong but you need to convince the reader that you have sufficient observations so please include a power analysis.
Line 296: did you explore any other time frames or metrics for change in activity and deviation than the rate of change from -3 to calving or -1? It would help the reader to understand if this was the optimal model after more data exploration or if this was the initial objective and other ways to explore activity data were not explored.
Line 350: Please consider a clearer way to show your models in a table as this is not easy to follow what was significant in the discussion and the results are brief.
Author Response
Dear reviewer
We would like to thank you for the time and effort you put into revising our manuscript. Please find the attachment file for our response.

Reviewer 3 Report
The study was scientifically sound and the article well-written. I believe it is very good that also results from studies that did not find relations, are published.
The authors studies the relation between prepartum behavioural data, automatically gathered with accelerometers, and postpartum hypocalcemia and hypomagnesemia. Many research groups are trying to predict disease from sensor data, but this specific relation was not yet studied.
As mentioned above, this relation was not yet studied, and although they did not find any relations, it is good to publish that, to prevent others from trying the same in the same way and not finding results.
The methodology was correctly chosen and the authors did use the correct controls (animals in the same conditions that did not develop hypocalcemia or hypomagnesemia).
The conclusions are consistent with the evidence and arguments presented, but they did not find the relations they expected.
The tables and figures clearly visualize the results and show that the differences between the groups were either very small or too variable to be significant.
Author Response
Dear reviewer
We would like to thank you for the time and effort you put into revising our manuscript. Please find the attachment for our response.
